# Effect of Apple Juice Enrichment with Selected Plant Materials: Focus on Bioactive Compounds and Antioxidant Activity

**DOI:** 10.3390/foods12010105

**Published:** 2022-12-25

**Authors:** Katarzyna Angelika Gil, Aneta Wojdyło, Paulina Nowicka, Paola Montoro, Carlo Ignazio Giovanni Tuberoso

**Affiliations:** 1Department of Life and Environmental Sciences, University of Cagliari, Cittadella Universitaria di Monserrato, S.P. Monserrato-Sestu km 0.700, 09042 Monserrato, Italy; 2Department of Fruit, Vegetable and Plant Nutraceutical Technology, Wrocław University of Environmental and Life Sciences, 37 Chełmońskiego Street, 51-630 Wroclaw, Poland; 3Department of Pharmacy, University of Salerno, Via Giovanni Paolo II, 132, 84084 Fisciano, Italy

**Keywords:** *Malus domestica* L., by-products, phenolic compounds, LC-PDA/MS QTof, UPLC-PDA, HPLC-ELSD, sensory evaluation

## Abstract

Using a multi-analytical approach, this paper aimed to investigate the effect of apple juice enrichment with *Arbutus unedo* and *Diospyros kaki* fruits, *Myrtus communis* berry extract, *Acca sellowiana,* or *Crocus sativus* flower by-products on both bioactive compounds content and antioxidant activity. Physico-chemical parameters, vitamin C, sugars, organic acids, total polyphenol content, antioxidant activity, and sensory attributes were evaluated. An LC-PDA/MS QTof analysis allowed for the identification of 80 different phenolic compounds. The highest polyphenol content (179.84 and 194.06 mg of GAE/100 g fw) and antioxidant activity (CUPRAC, 6.01 and 7.04 mmol of Fe^2+^/100 g fw) were observed in products with added *A. sellowiana* and *D. kaki*, respectively. Furthermore, the study showed a positive correlation between polymeric procyanidins and antioxidant activity (0.7646–0.8539). The addition of *A. unedo* fruits had a positively significant influence on the increment of vitamin C (23.68 ± 0.23 mg/100 g fw). The obtained products were attractive to consumers, especially those with 0.1% *C. sativus* flower juice, *M. communis* berry extract, and persimmon *D. kaki* fruits. The synergy among the different analytical techniques allowed us to obtain a complete set of information, demonstrating that the new apple smoothies were enriched in both different beneficial molecules for human health and in antioxidant activity.

## 1. Introduction

Apple trees (*Malus domestica* L. Borkh) are the most widely grown species in the genus *Malus* [1], and they are cultivated worldwide as fruit trees. Apple fruit consumption is widespread around the world, and it is on the market year-round. The apple market size is expected to increase by 420.04 million tons from 2021 to 2026, showing an accelerating growth of USD 10.12B, at a compounded average growth rate (CAGR) of 2.38%. The largest market share is in the Asia Pacific region (39.34%), and the US, Turkey, China, India, and Germany are expected to remain the largest markets for apple [2].

Apples are the main source of flavonoids in the diet of Americans and Europeans [3]. One kilogram of fruit contains 0.1 to 5.0 g of polyphenolic compounds, of which about 50% is flavan-3-ols (especially proanthocyanidins). The remainder of these biologically active structures are dihydrochalcones (including phloridzin and phloretin), phenolic acids (mainly chlorogenic acid), flavonols (quercetin glycosides) and, to a much lesser extent, anthocyanins [4,5]. Other valuable components of apples are pectin, a soluble fraction of dietary fibre (0.5–1.5% in relation to the weight of fresh fruit), and mineral compounds, including potassium, magnesium, and phosphorus. The health-promoting properties of *M. domestica* fruits and their products have been verified for a long time. In vitro and in vivo studies and clinical experience suggest comprehensive possibilities of using apples in the prevention and therapy of civilization diseases and a number of other diseases and dysfunctions of the body. Finnish researchers have found that apple consumption may reduce the incidence of asthma and type II diabetes [6]. Apples and their products were also clinically analysed for the prevention of overweight and obesity [7,8], as well as the prevention of cardiovascular disease [9]. The majority of apples are consumed fresh, while small parts are integrated into different food products, such as cookies, pastries, jams, and jellies, or processed into purées, concentrates, and juices [2,10]. Current healthy lifestyle trends place an important focus on the food industry. Consumer preferences for choosing ready- and easy-to-drink beverages make liquid or semi-liquid products desirable [11,12]. Juices and smoothies are increasingly popular ways of consuming fruits, and the industry is currently focusing maintaining the content of bioactive compounds. Apple juice, due to its long shelf-life and mass production, is the most popular juice used for processing in Europe, and the second most popular worldwide [13]. Apple juices, especially cloudy ones, are a rich source of natural antioxidants (e.g., polyphenols) [14] with a positive impact on human wellness, and which may be used in pharmaceutical or functional food preparation. Moreover, cloudy apple juice contains natural colloidal suspensions (e.g., proteins, pectins, and free amino acids), which have a positive influence on the stabilization of the colloidal system [15]. Due to all these aspects, apple juice is a well-known base for functional food preparation, mixing it with other liquid or solid ingredients to obtain formulation with improved physicochemical properties, a desired nutritional profile, and sensory acceptability [16]. Apple juice enrichment with plant bioactive compounds may be an excellent alternative for beverage development with a high impact on public health [17]. 

The consumption of beverages enriched with plant polyphenols has been reported to have potentially beneficial effects on human health [18]. However, the positive influence of these bioactive compounds depends on their nature, bioavailability, and the amount consumed [17]. Regardless, consumer desirability for innovative food products with beneficial functionality motivates researchers to investigate new plant materials, including plant by-products, that may potentially be used to develop new products for the food, beverages, and health industry [19,20]. For instance, Kanur et al. [21] pointed out in their review the importance of different parts of *Diospyros kaki* L. fruit, its nutritional value, and potential nutraceutical properties to develop functional foods. Altunkaya et al. [17] investigated the potential health effect of pomegranate peel extract on apple juice fortification; moreover, in their study, major antioxidant activity was observed in juice enriched with the highest selected concentration of by-product extract. On the other hand, Kolniak-Ostek et al. [18] evaluated the physico-chemical properties of cloudy apple beverages supplemented with apple leaves. Their results showed the increment of phytochemicals content after by-product addition, as well as their positive correlation with antioxidant activity. Biegańska-Mercik et al. [22] provided evidence of the positive influence of kale leaves addition to apple juice on photochemical composition and antioxidant activity, and assessed the effect of bergamot juice (a waste product) on the reduction in ascorbic acid and increase in the antioxidant activity of fruit juices (apple and apricot) [23]. Recently, Bayram and Sagdic [24] proved the beneficial influence of saffron microcapsules on the colour, antioxidant, and sensory properties of apple juices.

This paper aimed to investigate the effect of apple juice enrichment with specific plant materials (*A. unedo* and *D. kaki* fruits, *M. communis* berries, *A. sellowiana* and *C. sativus* flowers) on both bioactive compounds and antioxidant activity. For this purpose, a multi-analytical approach was applied to obtain a wide set of data useful to characterize such a complex mixture of vegetable products. LC-PDA/MS-QTof, UPLC-PDA, and HPLC-ELSD method were used for the qualitative and quantitative evaluation of the bioactive compounds. Spectrophotometric methods were used to evaluate colour parameters (L*, a*, b* and ΔE*), total polyphenol content, and antioxidant activity (CUPRAC, FRAP, ORAC, DPPH^●^, ABTS^●+^ assays); moreover, physico-chemical parameters and vitamin C were evaluated. Finally, a consumer evaluation was performed by measuring the quality of sensory attributes (colour, aroma, taste, consistency, desirability, and aroma type).

The plant materials used in this study to enrich apple juice and increase its value were selected for both their beneficial effects and generally recognized safe consumption. For this purpose, a detailed analysis of the scientific literature was conducted to confirm the safe usage of the chosen plant materials and their by-products. *A. unedo* fruits are eaten fresh and are well known for being processed mainly into alcoholic drinks, marmalades, jams, and jellies [25,26]; no toxic effects were reported [27]. *M. communis* berries are edible [28,29], and the most known application is the hydroalcoholic infusion of myrtle berries in order to obtain the famous Sardinian sweet liqueur [30]. *D. kaki* fruit is edible and highly beneficial for human health [31]. However, it is reported [32] that its overconsumption may cause some allergic reactions in certain individuals; thus, this aspect should be considered, and consumption should be sensible and limited for people with a potential allergy. *C. sativus* petals, to the best of our knowledge, have a limited use in the preparation of foodstuffs; they are mainly added as a garnish to dishes. Yet, in recent years, there has been an increasing interest in this by-product as a good source of compounds with beneficial properties [19,33] Preliminary in vitro studies [19,34] have confirmed a lack of cytotoxicity in the different studied extracts. *A. sellowiana* flowers are considered edible; the flower’s petals are added to salads and sweets and used as dish decorations [35]. However, for this product, only preliminary in vitro studies [20] are available, proving a lack of cytotoxicity and good antioxidant properties.

## 2. Materials and Methods

### 2.1. Chemicals and Standards

Acetonitrile for ultra-pressure liquid chromatography (UPLC, gradient grade), neocuproine, and ammonium acetate were purchased from Merck (Darmstadt, Germany), while copper (II) chloride and sodium acetate were purchased from Carlo Erba (Milan, Italy). Methanol, sodium acetate, phloroglucinol, sodium carbonate decahydrate, iron (II) sulphate heptahydrate, iron (III) chloride hexahydrate, rhamnose, fructose, sorbitol, glucose, sucrose, oxalic, citric, tartaric, malic, quinic, ascorbic, shikimic, fumaric, hydrochloric, formic, phosphoric, acetic acid and gallic, Folin–Ciocalteu reagent, Trolox (6-hydroxy-2,5,7,8-tetramethylchroman-2-carboxylic acid), TPTZ (2,4,6-tripyridyl-1,3,5-triazine), DPPH^•^ (2,2-diphenyl-1-picrylhydrazyl), ABTS (2,2′-azinobis-(3-ethylbenzthiazoline-6-sulfonic acid), AAPH (2,2′-azobis (2-amidino-propane) dihydrochloride), fluorescein disodium (FL), potassium persulfate, sodium and calcium chloride, disodium and dipotassium phosphate, starch from potato, *α*-amylase from porcine pancreas (type VI-8), 3,5-dinitrosalicylic acid (DNS), potassium sodium tartrate tetrahydrate, *p*-nitrophenyl-*α*-D-glucopyranoside (*p*NPG), *α*-glucosidase from *Saccharomyces cerevisiae* (type I), lipase (EC 3.1.1.3) from porcine pancreas (type II), *p*-nitrophenyl acetate, and 4-methylumbelliferyl oleate (4-MUO) were purchased from Sigma-Aldrich (Steinheim, Germany). Cyanidin-3-*O*-galactoside, cyanidin-3-*O*-glucoside, cyanidin-3-*O*-arabinoside, delphinidin-3-*O*-glucoside, malvidin-3-*O*-glucoside, peonidin-3-*O*-glucoside, petunidin-3*-O-*glucoside, phenolic acids (neochlorogenic, chlorogenic, caffeic, *p-*coumaric, ellagic), phloretin, (-)-epigallate, (-)-epicatechin, (+)-catechin, procyanidin B1 and B2, isorhamnetin-3-*O*-rutinoside, kaempferol-7*-O-*glucoside, kaempferol, myricetin-3-*O*-galactoside, myricetin-3-*O*-rhamnoside, myricetin, quercetin-4‘-*O*-glucoside, and quercetin-3*-O*-rhamnoside were from Extrasynthese (Genay Cedex, France). UPLC grade water, prepared by using an HLP SMART 1000 s system (Hydrolab, Gdańsk, Poland), was filtered through a 0.22 μm membrane filter immediately before use.

### 2.2. Plant Materials

Fresh apples (*Malus domestica* cv. Šhampion) were purchased at commercial maturity from the LA-SAD SP. Z.O.O (Borzęcin, Błędów, Poland) and persimmon fruits from the plantation “Melotto” (Villacidro, Sardinia, Italy). Other plant materials (*A. unedo* fruits, *M. communis* berries, *C. sativus,* and *A. sellowiana* flowers) were collected with random-block design sampling by professional pickers from plants growing under environmental conditions in Sinnai, Monte Arcosu, San Gavino Monreale, and Uta (Sardinia, Italy), respectively. The specimens were identified by Prof. Andrea Maxia (University of Cagliari, Italy), and voucher samples (number DISVA.ALI.05.2021, DISVA.ALI.06.2021, DISVA.ALI.09.2021, and DISVA.ALI.10.2021) were deposited at the Department of Life and Environmental Sciences of the University of Cagliari (Italy).

### 2.3. Apple Juice and Apple-Enriched Smoothies Production

The apple-based beverages production process included three main technology stages:(i)Apple juice production. Apple fruits were hand-washed in distilled water, cut into halves and arils, and ground in a Thermomix appliance (Vorwerk, Wuppertal, Germany) for 20 s with 0.2 mL of Pectinex Smash XXL enzyme per 1 kg of fruit. Then, the mashes were pressed in a hydraulic press (pilot plant laminar press; 15 tons of pressure) to obtain the juice.(ii)The processing of strawberry tree and persimmon fruits, myrtle berries, saffron and feijoa flowers. Strawberry tree fruits and feijoa flowers were frozen, freeze-dried, and homogenized into powder using a closed laboratory mill to avoid hydration (IKA 11A; BIOSAN, Vilno, Lithuania). Myrtle berries were double-extracted with ethanol 96% (1:1, *w*/*v*) under sonication for 30 min. Then, the extract was filtered using a strainer and concentrated by vacuum distillation (Büchi Rotavapor R-114, Switzerland) until complete alcohol elimination was achieved. Saffron flowers, the by-product obtained after the removal of the stigmas, were squeezed (manual press) to obtain juice, which was centrifuged, filtered using 0.45 μm cellulose acetate filter, frozen and freeze-dried. Persimmon fruits were grounded, heated at 80 °C in a Thermomix appliance (Vorwerk, Wuppertal, Germany), mashed, and reduced down in a blender (Symbio, Zelmer, Rzeszów, Poland) to a thin purée. After that, the purée was cooled and used to produce juices.(iii)Mixing semi-finished products. Apple juice (AJ) and other plant semi-products were mixed in appropriate proportions (*w*/*w*; 95:5 for *A. unedo* fruits (AJ+C5), *M. communis* berry extract (AJ+M5), *D. kaki* purée (AJ+P5) and *A. sellowiana* flowers (AJ+F5), and 99.99:0.01 (AJ+S01) and 99.95:0.05 (AJ+S05) for *C. sativus* flower juice) (Appendix A). Then, all products were heated to 100 °C, hot-filled in glass jars (135 mL), pasteurised (10 min at 90 °C), and cooled to 20 °C. The seven different products were analysed immediately after processing.

### 2.4. Physico-Chemical Analyses

All physico-chemical analyses were performed according to Wojdyło et al. [36], and all determinations were performed in triplicate. The dry matter was evaluated by the gravimetric method [37], and the results were expressed as g per 100 g of fresh weight (fw). The total soluble solids (TSS) content was determined by refractometer PAL-88S (Atago Rx 5000, Atago Co. Ltd., Tokyo, Japan) and expressed as °Brix [38]. Titratable acidity (TA) was determined by titration aliquots of the homogenate of fresh final products expressed as g of malic acid (MA)/100 g fw [39]. The total content of L-ascorbic acid (vitamin C) and ash content of the fresh final products were determined by the PN norms [40,41], and the results were expressed as mg or g, respectively, per 100 g of fw. 

### 2.5. Determination of Sugar and Organic Acids Content

The HPLC-ELSD and UPLC-PDA methods were used to determine the sugar content and organic acids, respectively, as described previously by Nowicka et al. [42]. All determinations were performed in triplicate, and the results were expressed as g/100 g dm of fw.

### 2.6. Colour Measurement

The colour parameters (L*, a*, b*, and ΔE*) were determined by reflectance measurement with a Colour Quest XE Hunter Lab colourimeter according to Nowicka et al. [43]. The samples were filled in a 1 cm cell, and L*a*b* values were measured against a white ceramic reference plate (L* = 93.92, a* = −1.03, b* = 0.52). The total colour change (ΔE*) was calculated following Wojdyło et al. [44]. Data were the mean of three measurements.

### 2.7. Identification and Quantification of Polyphenolic Compounds

The methods for the identification (LC-PDA/MS QTof) and quantitative (UPLC-PDA) analysis of polyphenols were performed as described previously by Wojdyło et al. [36]. The phenolic compounds were monitored at 280 nm (dihydrochalcones and flavan-3-ols), 320 nm (phenolic acids), 360 nm (flavonols), and 520 nm (anthocyanins). The results were expressed as mg per 100 g of fw.

### 2.8. Analysis of Polymeric Proanthocyanidins by Phloroglucinol Method

The analysis of polymeric procyanidins by the phloroglucinol method was performed according to Kennedy and Jones [45]. The results were expressed as mg per 100 g fw.

### 2.9. Determination of Total Phenolic Content (Folin–Ciocalteu Assay), Total Reducing Power (FRAP, CUPRAC Assays) and Free Radical Scavenging Activity (DPPH^•^, ABTS^•+^, ORAC Assays)

The total polyphenolic content (TP) was determined with a modified Folin–Ciocalteu method [46], and the results were expressed as mg of gallic acid equivalent (GAE) per 100 g of fw. The cupric ion-reducing antioxidant activity (CUPRAC) assay was performed according to the procedure of Bektaşǒglu et al. [47] with slight modifications, and the results were expressed as millimoles of Fe^2+^ per 100 g fw. FRAP, ORAC, ABTS^•+^, and DPPH^•^ assays were performed according to the procedure of Benzie and Strain [48], Ou et al. [49], Re et al. [50], and Tuberoso et al. [46], respectively. The obtained results of these four methods were expressed as millimoles of Trolox per 100 g fw. All the assays were measured spectrophotometrically in triplicate.

### 2.10. Consumer Evaluation of the Enriched Apple Smoothies

The sensory assessment of all products was carried out using a 5° hedonic scale with boundary indications: ‘I do not like it very much’ (1) to ‘I like it very much’ (5) [43,51]. The sensory tests were conducted in a sensory analysis laboratory designed according to ISO 8589:2009 standards. The laboratory was located at the Faculty of Biotechnology and Food Sciences, Wrocław University of Environmental and Life Sciences (Poland). The sensory evaluation sessions were conducted from 10 a.m. to 1 p.m. by 9 fully trained panellists between the ages of 25 and 45 years. All panellists received the same training to accustom them to the sensory attributes of all prepared products, and to understand the descriptors being used. The sensory evaluation of the juice and smoothies was performed using the 5-point hedonic scale (like very much—5, like—4, neither like nor dislike—3, dislike—2, and dislike extremely—1). The panellists scored the products for colour, aroma, taste, consistency, and desirability, and the results were produced with standard deviations. For the sensory evaluation, all products were labelled with codes, and the test was performed at room temperature. The products were served in a small, transparent plastic glass. After tasting each sample, the panellists neutralized their mouths.

### 2.11. Statistical Analysis

All data included in this study are presented as the mean value (*n* = 3) ± standard deviation. All statistical analyses were performed with Statistica version 7.0 (StatSoft, Krakow, Poland). Significant differences (*p* ≤ 0.05) between means were evaluated by a one-way ANOVA and Duncan’s multiple-range test. A correlation analysis was performed, and the evaluation of the statistical significance (*p* ≤ 0.05 and *p* ≤ 0.01) of observed differences was performed by using Spearman coefficients of correlation.

## 3. Results and Discussion

### 3.1. Physico-Chemical Parameters of the Apple Juice and Enriched Apple Smoothies

The main physico-chemical parameters (dry matter (DM), ashes, total soluble solids (TSS), total acidity (TA), pH, vitamin C, colour, sugars and organic acids content) were evaluated in all obtained products immediately after processing (Table 1 and Table 2). 

Statistically significant differences (*p* ≤ 0.05) were found among the analysed mixed plant material products, showing the influence of their different chemical compositions. The highest DM content was detected in AJ+F5 and AJ+C5 (17.59 and 16.97 g/100 g fw, respectively), while the lowest values of dry matter (13.54 and 13.88 g/100 g fw) were observed in 100% apple juice and AJ+S01, respectively. The highest values of ashes were observed in the smoothie with additional 5% feijoa flowers (0.42 g/100 g fw), while the lowest were in the product with additional 0.1% dry saffron flower juice (0.13 g/100 g fw). The highest content of soluble solids (TSS) was found in AJ+F5 and AJ+C5 (16.00 and 16.10 °Brix), while the lowest was detected in 100% apple juice (13.20 °Brix). The TSS is a parameter that largely determines the final dry matter content [42] and depends on the content of soluble compounds such as dyes, tannins, and non-volatile organic acids (citric, tartaric, or malic), and principally on the total sugar content [52]. Thus, TSS content is usually higher in strongly coloured fruits containing more sugars and acids. Also observed in this study was a relationship between the content of TSS in the analysed final products and their DM content (r = 0.9795, *p* ≤ 0.01, Appendix A). This is consistent with the findings reported by Vidrih et al. [53]. The DM content and the TSS content not only depend on the cultivar, but may also be influenced by many other factors, such as the degree of fruit dehydration, harvest time, climatic and agricultural conditions, and an increase in the insoluble solids content of the fruit during maturation [54].

The highest content of titratable acidity (TA) was determined in AJ+F5 and AJ+C5 (0.49 and 0.50 g of MA/100g fw), while the lowest was in AJ (0.42 g of MA*/100g fw). TA was strongly correlated with malic acid, total organic acids, and total hydroxybenzoic acids (*p* ≤ 0.01, Appendix A). As confirmed by Oliveira et al. [55], A. unedo fruits are characterized by a high content of organic acids, while A. sellowiana flowers have never been investigated before for their organic acid content, but their profile and content might be similar to that of feijoa fruits [56]. Therefore, these two semi-products proved responsible for the acidity of the enriched apple smoothies. The low total acidity is important in terms of technological preservation because of possible problems in the pasteurisation process during beverage preparation at a pH below or above 4.6 [44]. TSS and TA parameters, as well as the ratio between them, are commonly used by the juice industry as quality control indicators. The obtained smoothies showed values of TSS/TA that were generally lower than the AJ, except AJ+F5, which was slightly higher (32.65). According to Jaros et al. [57], TSS/TA ratio is the most important parameter helping to predict consumer preferences for cloudy apple juices; some consumers generally prefer sweeter juices, with higher ratios of TSS/TA [57]. Therefore, as described by Konic-Ristic et al. [58] industrial processors of commercial juices have different requirements regarding acidity in the raw materials because low ratios are a good indicator for prolonging fruit quality during storage. In addition, the obtained results show a significant difference in the values of these important parameters that influence not only the sensory quality, but also the colour intensity and microbiological stability of final products [59]. The pH of all analysed products ranged from 3.31 to 4.00; it was highest for AJ+F5 and lowest for 100% apple juice. Analysing the content of vitamin C (ascorbic acid) showed that the investigated enriched apple smoothie with additional 5% strawberry tree fruits contained a much higher amount of this vitamin (23.68 mg/100 g fw) than the other investigated beverages (0.64–0.95 mg/100 g fw). The high content of vitamin C in AJ+C5 was due to the presence of A. unedo fruits, which are a rich source of this vitamin, according to Vidrih et al. [53].

Another fundamental quality parameter analysed in all the prepared products was the colour, and the four main colour determinants (lightness L*, redness a*, yellowness b*, and total change in colour ΔE*) were studied. The value of the L* parameter in the analysed products ranged from 29.25 to 54.92. The highest value of parameter L* was detected in AJ+C5, followed by AJ+P5 (L* = 54.75). In turn, the lowest value of parameter L* was detected in AJ+M5. Generally, it was observed that when semi-products (saffron flower juice, myrtle berry extract, and feijoa flowers) were added, the juice became darker; the exceptions were AJ+P5 and AJ+C5, which were brighter than the control juice. The value of a* parameter ranged from 1.49 to 12.36. The most intense red colour was detected in AJ+C5, while the lowest value of a* parameter was observed in AJ. Furthermore, it is worth noting that each semi-product addition to apple juice increased the a* value, which resulted in the investigated apple smoothies showing a more intense reddish colour. On the other hand, the value of b* ranged from −1.48 to 24.99. The most intense yellow colour was detected in AJ+C5, while the lowest b* value was detected in AJ+M5. It is also worth noting that the value of the b* parameter continued to change when saffron flower juice, myrtle berry extract, feijoa flowers, and strawberry tree or persimmon fruits were added. It was observed that the control juice was less yellow after the addition of plant material components; the exceptions were AJ+P5 and AJ+C5, which appeared to be more yellow than AJ. In addition, the different colour of the analysed products is related to the nature of the additional plant materials. Persimmon and strawberry tree fruits are characterised by red-orange skin and pulp; therefore, the a* and b* values were higher in beverages containing these semi-products than in the control juice; however, high a* and b* values in final products with D. kaki and A. unedo may be due to a high content of carotenoids [60,61]. Moreover, feijoa flowers, saffron flower juice, and myrtle berries are characterised by a blue-violet colour; thus, the a* value was higher but the b* value was lower in all smoothies containing these semi-products than in the control apple juice. This variation could be due to the nature of the pigments in these plant material cultivars, especially because of the anthocyanins content, which resulted in a darker colour. In addition, these differences in the results might be attributed to the differences in anthocyanin composition [62] present in the added plant materials. As described by Koponen et al. [63], in nature, cyanidin has more redness than delphinidin derivatives. Finally, the parameter ΔE*, describing the human eye’s ability to discriminate between the colours of two different products, is also important from the perspective of the processing industry. It is known that consumers can distinguish the colour of two different samples when values of ΔE* are higher or equal to 5 [59]. The values of the ΔE* parameter in the analysed products immediately after processing ranged from 1.80 to 28.52. The highest value of ΔE* was detected in AJ+M5, while the lowest value of ΔE* was observed in AJ+S01.

### 3.2. Sugar and Organic Acids Content of the Apple Juice and Enriched Apple Smoothies

Investigation of the sugar and organic acid content (Table 2) confirmed that mixing various plant materials into apple juice allowed for enriching the smoothies with specific sugars and organic acid.

The total sugar content in all the analysed products ranged from 8.10 to 14.85 g/100 g fw. The highest total sugar content was found in AJ+C5, while the lowest was in the control juice. It is worth noting that enrichment in all the used semi-products increased the total sugar content of apple juice. Fructose and glucose were abundant sugars, while the sorbitol and sucrose contents were much lower. However, in analysing every sugar, some interesting results were observed. The fructose and glucose content in all beverages were in the range of 6.73–11.72 and 0.97–2.57 g/100 g fw, and the highest content was detected in AJ+C5 and AJ+M5, respectively. Moreover, the 5% addition of persimmon purée did not influence the fructose content of the smoothie, and it was not significantly different compared to AJ. Next, the sorbitol and sucrose content in all products ranged from 0.07 to 0.16 and 0.23 to 0.57 g/100 g fw, respectively. The highest content of sorbitol and sucrose was detected in AJ+C5, while the lowest sucrose content was found in AJ+M5. It is noteworthy that the 0.1% addition of saffron flower juice did not influence the sorbitol content compared to the control juice. The presence of sorbitol can have a positive influence on final product preparation because it is non-cariogenic; therefore, it helps protect against tooth decay. Moreover, it slows the rise of blood glucose and the insulin response connected to the ingestion of glucose. Hence, it can be used as a sugar alternative for people with diabetes [52]. It should be noted that the final products were rich in fructose, which has better metabolic properties than glucose and sucrose due to its lower glycaemic index. Moreover, regarding sugar composition in the investigated plant materials, it is known that fructose and glucose are sweeter than sucrose [36], and fructose has higher relative sweetness than glucose [64]. Therefore, the consumers’ perception of sweetness in the analysed final products is, in all likelihood, mostly related to their content of fructose and glucose. Mixing different plant material semi-products may be a good way to correct overly sour or sweet tastes, which would lead to obtaining an attractive product, as in the present study. Moreover, the impact of sugar-containing foods varies according to food class. Given that excessive dietary sugar intake may be associated with an increased risk of cardiovascular and metabolic diseases, it is, therefore, necessary to limit the daily consumption of food products containing particular quantities of total sugar content to within the healthy recommended sugar limits. Finally, the quality of foods should be assessed not only based on their sugar content, but also on their micronutrient and dietary fibre contents [65].

Eight organic acids were identified and quantified in the analysed beverages (Table 2). Generally, the total organic acid content in all analysed products ranged from 0.94 to 1.57 g/100 g fw. The highest total organic acid content was determined in AJ+F5, while the lowest was in the control juice, which is the simplest version among all beverages. This suggests that this by-product was the most acidic out of all the additional components. Moreover, in analysing every organic acid, some interesting results were observed. Quinic and malic acids were the most abundant organic acids (0.44–0.84 and 0.48–0.58 g/100 g fw, respectively) among all apple-based products, while other acids (oxalic, citric, tartaric and shikimic) were detected in much lower quantities (0.01–0.29 g/100 g fw). Additionally, the presence of ascorbic and fumaric acids was confirmed. Tartaric acid was found only in AJ and AJ+P5, while the oxalic and citric acids were detected in significantly higher amounts in AJ+F5 than in other smoothies. Interestingly, the total amount of organic acids strongly positively correlated with the amount of hydroxybenzoic acids (r = 0.9788, *p* ≤ 0.05, Appendix A). The organic acids present in fruits possess pro-health benefits. These compounds have the power to stimulate the secretion of digestive enzymes, as well as to regulate the proper chemical reactions of the body [66]. Moreover, the presence of particular organic acids guarantees specific taste, flavour, and aroma, and contributes to stabilising and preserving foodstuffs [67].

### 3.3. Identification and Quantification of Phenolic Compounds in the Apple Juice and Enriched Apple Smoothies

The LC-MS analysis of all final products revealed the presence of 80 different phenolic compounds belonging to six subclasses: anthocyanins, hydroxybenzoic and hydroxycinnamic acids, dihydrochalcones, flavan-3-ols (monomers, dimers, trimer, and polymeric procyanidins) and flavonols (Figure 1, Appendix A). Moreover, these compounds varied between all obtained beverages. Regarding each analysed product, the LC-MS metabolic profiles highlighted the presence of a large group of polyphenols, precisely: 17 (AJ), 25 (AJ+S01), 29 (AJ+S05), 34 (AJ+M5), 35 (AJ+F5), 19 (AJ+K5), and 38 (AJ+C5). 

The total concentration of these compounds, as measured by the UPLC-PDA method, ranged from 131.93 to 274.40 mg/100 g fw for AJ < AJ+S05 < AJ+S01 < AJ+M5 < AJ+K5 < AJ+C5 < AJ+F5, respectively. Statistically significant differences (*p* ≤ 0.05) were observed between the obtained results (Figure 1, Appendix A). The content of polyphenolic compounds in pure apple juice was lower than in smoothies because the addition of each semi-product to apple juice increased the phenolic content in the obtained products.

Anthocyanins are a subclass of polyphenols that were detected in almost all the investigated products, except AJ and AJ+K5. Therefore, the presence of these compounds in the analysed smoothies was due to the enrichment with the plant material semi-products. Apple smoothie enriched with A. unedo fruits showed two delphinidin derivatives, from which the most abundant was cyanidin-3-*O*-galactoside (1.04 mg/100 g fw), while cyanidin-3-*O*-arabinoside was found in much lower amounts (0.04 mg/100 g fw, respectively). Considering the results of the smoothie with the addition of M. communis berries extract, delphinidin-3-*O*-glucoside, cyanidin-3-*O*-glucoside, and malvidin-3-*O*-glucoside (10.55, 3.32 and 17.05 mg/100 g fw, respectively), three main compounds were responsible for the red-purple colour. The values of the other three anthocyanins detected in AJ+M5 ranged from 0.04 (peonidin-3-*O*-glucoside) to 0.83 mg/100 g fw (petunidin-3-*O*-glucoside). Furthermore, in products containing floral by-products (AJ+S01, AJ+S05, and AJ+F5) a few different anthocyanins were found. In the product with added saffron flower juice, delphinidin-3,5-*O*-diglucoside and delphinidin-3-*O*-glucoside (0.17–1.17 and 0.37 mg/100 g fw, respectively) were detected. In turn, in the smoothie with added feijoa flowers, cyanidin-3-*O*-glucoside (3.69 mg/100 g fw) was found. The presence of anthocyanins is valuable because they have a wide range of biological activities, such as antioxidant, cardioprotective, anti-inflammatory, antitumor, and eye function properties [68].

Hydroxybenzoic acids and their derivatives were found in smoothies enriched with myrtle berry extract, feijoa flowers, persimmon, and strawberry tree fruits, while in pure apple juice and with added saffron flower juice, none of these compounds were detected. The highest concentration of this subclass was detected in products enriched in A. unedo fruits and A. sellowiana flowers (29.28 and 28.37 mg/100 g fw, respectively). AJ+C5 was the most abundant in theogallin (23.10 mg/100 g of fw), while other compounds (gallic acid glucosides, galloyl glucosides, gallic acid 4-*O*-β-D-glucopyranoside, galloyl and digalloyl shikimic acids, digalloquinic acids, and strictinin ellagitannin) were detected in much lower amounts between 0.03 and 2.06 mg/100 g fw. Regarding AJ+F5, the most representative hydroxybenzoic derivative was castalagin and ellagitannin II (7.87 and 9.34 mg/100 g fw), while other compounds (casuarinin, ellagitannin IV, nilocitin, casuarinin, ellagic acid and its two pentosides (arabinoside and xyloside) and methyl ellagic acid) were present in lower amounts (0.26–3.56 mg/100 g fw). Moreover, in AJ+M5 and AJ+K5, some amounts of these compounds were detected. The first smoothie was rich in two galloyl HHDP-glucoses (c.a. 2.50 mg/100g fw), and other derivatives (ellagitannin I and III) were found in amounts between 0.13 and 1.50 mg/100 g fw. The last beverage in which hydroxybenzoic acid derivatives were found was AJ+K5. Two galloyl glucosides were found in this smoothie in amounts of 1.10 and 1.70 mg/100 g fw, respectively.

Hydroxycinnamic acids were the next group of polyphenols identified in all the analysed products in a range between 6.58 and 8.42 mg/100 g fw. The most abundant beverage in this subclass of polyphenols was AJ+C5, while the poorest was pure apple juice. All five hydroxycinnamic acids (neochlorogenic, chlorogenic, caffeic, p-coumaric, and p-coumaroylquinic acid) are the typical compounds found to occur in apple fruits, which is in line with the previous findings [5]. Chlorogenic acid was present in all products in the highest quantities (5.00 to 6.62 mg/100 g fw), while values of the other acids ranged between 0.10 and 1.01 mg/100 g fw. The obtained results showed that each additional component enriched apple juice with hydroxycinnamic acids.

Two dihydrochalcones (phloretin-2′-*O*-xyloglucoside and phloridzin) were found in the present study in all beverages. These compounds were present in a total quantity ranging from 4.32 to 5.19 and mg/100 g dm. Dihydrochalcones, principally phloretin, have the power to increase the adhering action of bioactive components to the surface of lipids, changing the bipolar potential of the lipid bilayer. Moreover, the presence of these compounds helps the inhibition of active glucose transporters into SGLT1 and SGLT2 cells, as well as various urea transporters. Additionally, they have strong antioxidant activity [69].

Flavan-3-ols (monomers, dimers, trimer, and polymeric procyanidins) were the major subclass in the analysed products. The highest total amount of these compounds was detected in AJ+F5 (223.21 mg/100 g fw) and in AJ+K5 (255.59 mg/100 g fw). In turn, the lowest total amount of these compounds was in AJ+S05 (100.27 mg/100 g fw). Monomers ((+)-catechin and (-)-epicatechin), and dimers (procyanidin B1 and B2) were detected in quantities ranging from 1.44 to 7.94 mg/100 g fw and 1.09 to 11.25 mg/100 g fw, respectively. The peculiarity of the smoothie containing strawberry tree fruits lay in the presence of procyanidin B3 (0.06 mg/100 g fw). Moreover, dimers were detected among all juices in a greater amount than trimer (procyanidin C1; 0.40–0.80 mg/100 g fw). The characterisation of the total flavan-3-ols subclass, besides monomers, dimers, and trimer, was also carried out for polymeric procyanidins (PP). These compounds were dominant in all the analysed juices (54–89% of total phenolic compounds), and the juices that were richest in them were AJ+F5 and AJ+K (209.99 and 242.68 mg/100 g fw, respectively). The exception was the product with 0.5% added saffron flower juice, which showed the lowest amount of these compounds (88.69 mg/100 g fw). In vitro and in vivo studies have confirmed the presence of (-)-epicatechin [70], a molecule that allows insulin synthesis stimulation and increases the level of cAMP in β cells of the pancreas, which increases the secretion of this hormone. Moreover, the transformation of proinsulin into insulin is more effective; thus, insulin levels in the blood are higher. Appendix A also reports the results of the degree of polymerization (DP; the number of flavan-3-ol units), which are responsible for the physicochemical properties of PP. The DP of the AJ polymeric fraction was 3.53, while in other beverages it was higher (3.64–4.78). The exception was AJ+S05, in which DP was 3.28.

Our analysis revealed statistically significant differences (*p* ≤ 0.05) in flavonols concentrations in the analysed products. The content of flavonols ranged from 1.63 (AJ) to 22.72 (AJ+S05) mg/100 g fw. The high value of these compounds was also detected in AJ+M5 (22.61 mg/100 g fw)). In all beverages were found quercetin-3-*O*-galactoside, quercetin-3-*O*-glucoside, and quercetin-3-*O*-rhamnoside in amounts ranging from 0.57 to 1.28, 0.10 to 0.40, and 0.50 to 1.52 mg/100 g fw, respectively. Furthermore, in all products were present two flavonol-*O*-pentosides, such as quercetin-3-*O*-arabinoside and -xyloside (0.14–0.27 and 0.32–0.74 mg/100 g fw), respectively. All of the above compounds were presented in pure apple juice, but it was noticed that each additional component increased the amount of these compounds in the enriched apple smoothies. Moreover, another quercetin-pentoside was found only in AJ+F5 (0.39 mg/100 g fw).

The products containing saffron flower juice (AJ+S01 and AJ+S05) were characterized by the presence of kaempferol derivatives in amounts between 0.04 and 2.13 and 0.15 and 11.84 mg/100 g fw, respectively. Kaempferol-3-*O*-sophoroside proved to be the major representative among all detected kaempferol derivatives. Isorhamnetin-3,7-*O*-digalactoside and -diglucoside and isorhamnetin-3-*O*-rutinoside were the other particular compounds detected in both AJ+S01 and AJ+S05 in a range between 0.17 and 0.24 and 0.20 and 1.32 mg/100 g fw, respectively. Additionally, c.a. 0.21 mg/100 g fw of isorhamnetin-3-*O*-sophoroside and isorhamnetin-3-*O*-glucoside were found only in AJ+S05. Moreover, small amounts (0.06 mg/100 g fw) of quercetin-3,7-*O*-digalactoside were found in AJ+S05, while quercetin-3,7-*O*-diglucosidase was detected in both products containing saffron flower juice (0.35 and 2.26 mg/100 g fw, respectively), as well as in much lower quantities in AJ+F5 (0.09 mg/100 g fw). 

Aside from the above-mentioned quercetin-pentoside, particular flavanols in AJ+F5 were kaempferol-3-*O*-galactoside, kaempferol-hexoside, quercetin, quercetin, and kaempferol (0.53, 0.21, 0.05, and 3.78 mg/100 g fw, respectively). On the other hand, AJ+M5 was characterized by the presence of myricetin galactoside-gallate, myricetin-3-*O*-arabinoside, and myricetin (1.75, 1.18 and 0.33 mg/100 g fw). 

In turn, only in AJ+C5 were found quercetin galloylhexose, quercetin derivative II, and myricetin-3-*O*-xyloside (0.03–0.27 mg/100 g fw). Furthermore, myricetin-3-*O*-galactoside, -glucoside and -rhamnoside were found in the smoothies enriched in 5% myrtle berries extract (9.85, 0.50 and 5.88 mg/100 g fw, respectively) and strawberry tree fruits (0.33, 0.25 and 0.02 mg/100 g fw, respectively), while 0.09 and 0.21 mg/100 g fw of quercetin derivative I was detected in AJ+F5 and AJ+C5, respectively.

It is noteworthy that each additional component had a positive influence on the phenolic composition of the enriched beverages; thanks to each particular plant matrix, apple juice became a more desirable nutraceutical product. 

### 3.4. Antioxidant Activity of the Obtained Apple Juice and Enriched Apple Smoothies

The total polyphenol (TP) content (Folin–Ciocalteu method) and antioxidant activity (CUPRAC, FRAP, ORAC, ABTS^•+^ and DPPH^•^) of all the investigated products were evaluated, as presented in Table 3, and they showed similar trends between the analysed beverages. Generally, all newly enriched apple smoothies showed stronger antioxidant activity compared to the control juice. An exception was the antioxidant activity of AJ+C5 measured by ORAC (2.10 mmol Trolox/ 100 g fw). The highest antioxidant activity measured by the FRAP, ORAC, ABTS^•+^, and DPPH^•^ methods was determined in AJ+F5 (1.77, 3.51, 2.21, and 1.26 mmol Trolox/100 g fw), while measured by the CUPRAC method was observed in AJ+P5 (7.04 mmol Fe2+/100 g fw). In contrast, beverages enriched with 0.5% saffron flower juice showed the lowest antioxidant potency (0.65, 0.50, and 0.57 mmol Trolox/100 g fw for FRAP, DPPH^•^, and ABTS^•+^) compared with other new smoothies. It is worth noting that, in the case of antioxidant measurement using the CUPRAC, FRAP, ABTS^•+^, and DPPH^•^ methods, significantly higher values were noted in products enriched with 0.1% saffron flower juice than in enriched apple juice, in which the amount of this by-product was higher (0.5%).

It was observed that the obtained results of the antioxidant activity of all products were positively correlated with the amount of TP dosed by HPLC-PDA and the Folin–Ciocalteu assay (r = 0.6275–0.9086). The correlation was statistically significant between TP by the Folin–Ciocalteu assay and CUPRAC, FRAP, DPPH^•^, and ABTS^•+^ (*p* ≤ 0.05, *p* ≤ 0.01, *p* ≤ 0.05, *p* ≤ 0.05, and *p* ≤ 0.01 respectively), and between TP dosed by HPLC-PDA and FRAP and ABTS^•+^ (*p* ≤ 0.01 and *p* ≤ 0.05, respectively) (Appendix A). Regarding the contribution of single polyphenolic classes, the most important significant correlation was observed between the total antioxidant activity and the polymeric procyanidins, which was significant at *p* ≤ 0.05 for the CUPRAC, FRAP, DPPH^•^, and ABTS^•+^ assays (Appendix A). This finding confirms that, besides the total content of phenolic compounds, the type of polyphenols also plays a very important role in antioxidant activity, as previously reported [52,71]. The total antioxidant capacity of food is the result of the activity of a number of compounds with diverse chemical structures and mutual interactions (synergism, and/or antagonism) [72]. It is generally recognized that the main antioxidants of plant materials are polyphenols and vitamins, including ascorbic acid [3,73]. The conducted study confirms the view prevailing in the literature that there is a strong relationship between the concentration of polyphenols and the antioxidant activity of plant raw materials and their products. The diversity of the structure of polyphenolic compounds plays a decisive role in shaping these properties. In the case of the obtained products, the most important from the perspective of pro-health properties are the content of compounds from the group of flavan-3-ols (including monomeric, dimeric, oligomeric forms, and polymers of proanthocyanidins), and anthocyanins. These compounds are characterized by high biological activity and are an important component in the diet of people suffering from chronic non-communicable diseases, especially heart and blood vessel diseases or obesity. Thus, juices or smoothies containing health-promoting compounds are able to prevent or inhibit the progression of degenerative diseases caused by oxidative stress [74]. In this context, it can be concluded that the developed products are promising in terms of the possibility of their use in the prevention of chronic diseases and as functional products with programmed health-promoting properties. 

### 3.5. Sensory Evaluation of the Obtained Apple Juice and Enriched Apple Smoothies

The sensory results obtained by the trained panel were grouped according to complex sensory properties: colour, aroma, taste, consistency, and desirability (Figure 2, Appendix A. The study showed that almost all beverages were attractive in terms of colour discrimination (≥3.00). However, the highest score (3.90 and 3.85) was obtained in juice with additional strawberry tree fruits (AJ+C5) and myrtle berry extract (AJ+M5), respectively, while the lowest score (1.60) was evaluated in juice containing feijoa flowers (AJ+F5). A good value of colour was also observed in pure AJ (score 3.40) and AJ+S01 (score 3.25). Although no statistically significant correlation (*p*≤ 0.5, Appendix A) was found with either colour parameters (L*, a*, b*, and ΔE*) or specific compounds (e.g., anthocyanins), the most accepted product was the reddest (AJ+M5) and the most yellow (AJ+C5) of all the analysed products.

According to consumers, the following beverages had the best aroma: AJ (score 4.60) and AJ+C5 (score 4.10), while AJ+F5 had the worst aroma (score 2.15). For the taste value, the best products (≥4.00) were: AJ+S01, AJ+P5, and AJ. The worst-tasting product (score 2.15) was the apple juice enriched with feijoa flowers. A significant correlation at *p* ≤ 0.01 (0.9117) was observed between aroma and taste. The lack of acceptance of this smoothie may be due to the high level of pH (Table 1) and the high amount of citric and oxalic acids (Table 2), because all these parameters strongly negatively correlated with the desirability (*p* ≤ 0.01, Appendix A). 

According to consumers, the consistency of the smoothies was very similar compared to the AJ (2.90), ranging from 2.80 to 3.00. The highest consistency evaluation score (3.00) was in AJ+M5 and AJ+P5, while the lowest score (2.80) was in product AJ+S05. This product resulted in more fluid than the typical smoothies; thus, the enrichment in 0.5% saffron juice did not provide a semi-liquid consistency, so it did not meet the strict requirements for smoothies. Finally, according to consumers, the best product (desirability score ≥ 4.10) besides AJ was AJ+S01. No statistically significant correlation (*p* ≤ 0.5, Appendix A) was found to justify the consumers’ appreciation for this smoothie compared to the others, although a low amount of total organic acid was observed (Table 2). This relationship was also confirmed by Nowicka et al. [43], who studied the sensory properties of sour cherry puree smoothies mixed with apple, pear, quince, and flowering quince juices. The attractiveness of products with additional saffron flower juice is interesting because this extract contains bioactive molecules that can benefit human wellness [19]. In contrast, smoothies containing feijoa flowers resulted as unacceptable, probably due to their brown colour and unpleasant solid parts, resembling rotten vegetables. However, due to their significant bioactive potential [20], feijoa flowers can be a good raw material, enriching the quality of the final product. A more appropriate technological process for preparing the feijoa plant material would probably make it more acceptable to the consumers. 

Finally, the consumers were asked about their preferred packaging for the analysed beverages, with the proposed packaging being a plastic bottle, Tetra Pak^®^ brick with a straw, glass bottle, glass jar, and pouch pack. In general, the most desirable were glass and plastic bottles.

## 4. Conclusions

This multi-analytical study showed that mixing apple juice with myrtle berry extract, saffron and feijoa floral by-products, or strawberry tree and persimmon fruits contributes to the enrichment of the final product in bioactive compounds. It also influences the sensory properties of the obtained apple products. Smoothies with myrtle berries extract contained all the phenolic compounds classes (anthocyanins, hydroxybenzoic and hydroxycinnamic acids, flavan-3-ols, flavonols, and dihydrochalcones). Smoothies with the addition of feijoa flowers and persimmon fruits were characterized by a high polyphenolic content. The enrichment of apple juice with *A. unedo* fruits had a significant positive influence on the increment of vitamin C. Strawberry tree fruits and feijoa flowers enriched new products in sugars and organic acids. Regarding antioxidant activity, the highest values were observed in products containing persimmon fruits and feijoa flowers. Generally, the new apple smoothies were appreciated by consumers, except the one obtained with feijoa flowers. However, feijoa flowers are a very good raw material rich in the bioactive compound, which, with the use of better technological processing, may result in an interesting ingredient in the preparation of smoothies. Finally, exploitation of the plant products and by-products used for this experimentation could have a positive input on local industry development and marketing.

## Figures and Tables

**Figure 1 foods-12-00105-f001:**
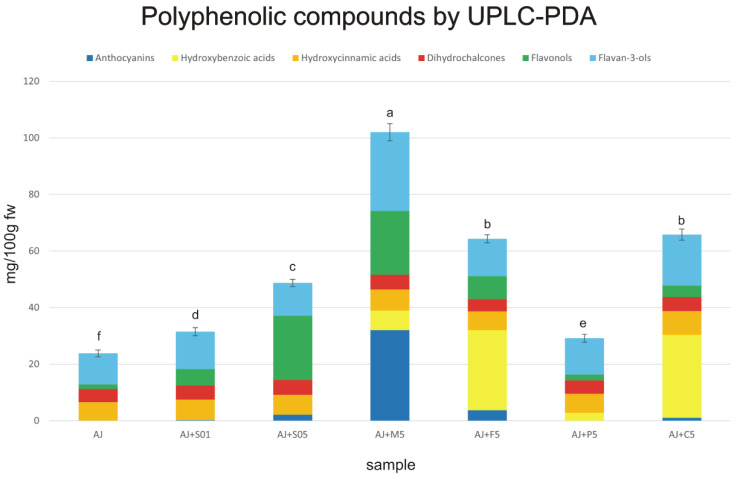
Quantification of phenolic compounds by UPLC-PDA method (mg/100 g fw) in apple juice and enriched apple smoothies. Mean values of the total polyphenolic compounds with different letters (a–f) are significantly different (homogenous groups) at *p* ≤ 0.05.

**Figure 2 foods-12-00105-f002:**
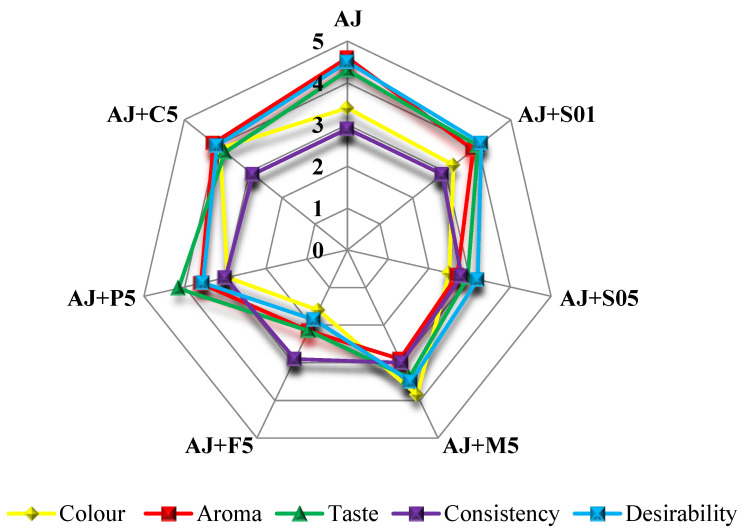
Consumer evaluation of enriched apple juices (5° hedonic scale). Sensory evaluation was carried out in analysed juices immediately after processing. The presented values are the average of all evaluations. AJ—apple juice; AJ+S01—apple juice + 0.1% saffron flower juice; AJ+S05—apple juice + 0.5% saffron flower juice; AJ+M5—apple juice + 5% myrtle berry extract; AJ+F5—apple juice + 5% feijoa flowers; AJ+P—apple juice + 5% persimmon purée; AJ+C—apple juice + 5% strawberry tree fruits.

**Table 1 foods-12-00105-t001:** Physico-chemical composition and colour parameters of apple juice and enriched apple smoothies.

Parameter	Sample						
	AJ	AJ+S01	AJ+S05	AJ+M5	AJ+F5	AJ+P5	AJ+C5
Dry matter (g/100 g fw)	13.54 ± 0.00 f	13.88 ± 0.07 e	14.48 ± 0.07 d	14.85 ± 0.09 c	17.59 ± 0.01 a	14.49 ± 0.31 d	16.97 ± 0.03 b
Ashes (g/100 g fw)	0.28 ± 0.04 cd	0.23 ± 0.00 d	0.34 ± 0.03 cd	0.33 ± 0.02 c	0.53 ± 0.04 a	0.28 ± 0.02 c	0.31 ± 0.00 b
Total soluble solids (TSS, °Brix)	13.20 ± 0.00 g	13.50 ± 0.02 f	14.10 ± 0.03 d	14.60 ± 0.01 c	16.00 ± 0.01 b	13.80 ± 0.02 e	16.10 ± 0.00 a
Total acidity (TA, g of MA ^#^/100 g fw)	0.42 ± 0.01 c	0.44 ± 0.01 bc	0.45 ± 0.00 b	0.45 ± 0.01 b	0.49 ± 0.01 a	0.44 ± 0.03 bc	0.50 ± 0.01 a
TSS/TA	32.20	30.68	31.33	31.74	32.65	30.00	31.57
pH	3.31 ± 0.03 e	3.52 ± 0.01 d	3.58 ± 0.02 c	3.68 ± 0.01 b	4.00 ± 0.02 a	3.59 ± 0.01 c	3.56 ± 0.04 cd
Vitamin C (mg/100 g fw)	0.90 ± 0.01 b	0.64 ± 0.01 c	0.65 ± 0.01 c	0.65 ± 0.01 c	0.95 ± 0.01 b	0.68 ± 0.04 c	23.68 ± 0.23 a
Colour parameters ^§^							
L*	50.54 ± 0.03 b	49.60 ± 0.05 c	44.13 ± 0.06 d	29.25 ± 0.00 f	37.29 ± 0.13 e	54.75 ± 0.06 a	54.92 ± 0.21 a
a*	1.49 ± 0.00 g	2.74 ± 0.02 f	6.11 ± 0.03 d	7.78 ± 0.00 b	7.30 ± 0.06 c	2.83 ± 0.02 e	12.36 ± 0.05 a
b*	16.43 ± 0.05 c	15.53 ± 0.02 d	13.88 ± 0.01 e	−1.48 ± 0.00 g	8.75 ± 0.03 f	18.13 ± 0.12 b	24.99 ± 0.02 a
ΔE*	-	1.80 ± 0.00 f	8.30 ± 0.02 d	28.52 ± 0.03 a	16.38 ± 0.11 b	4.73 ± 0.03 e	14.51 ± 0.02 c

Data are given as mean ± standard deviation (*n* = 3). Mean values within a column with different letters (a–g) are significantly different (homogenous groups) at *p* ≤ 0.05. ^#^ MA = malic acid. ^§^ STD parameter used for colour analysis is AJ. L* is lightness; a* indicates red for positive value and green for negative value; b* indicates yellow for positive value and blue for negative value; ΔE* is the total colour difference.

**Table 2 foods-12-00105-t002:** Sugar and organic acid content in apple juice and enriched apple smoothies.

Parameter	Sample						
	AJ	AJ+S01	AJ+S05	AJ+M5	AJ+F5	AJ+P5	AJ+C5
Sugar content (g/100 g fw)							
Fructose	6.73 ± 0.23 d	6.77 ± 0.15 d	9.34 ± 0.10 c	11.47 ± 0.05 a	10.93 ± 0.42 b	11.02 ± 0.01 b	11.72 ± 0.22 a
Sorbitol	0.07 ± 0.00 d	0.07 ± 0.01 d	0.10 ± 0.01 c	0.12 ± 0.02 bc	0.11 ± 0.00 c	0.14 ± 0.01 ab	0.16 ± 0.02 a
Glucose	0.97 ± 0.05 e	1.02 ± 0.10 e	1.33 ± 0.00 d	2.57 ± 0.01 a	2.03 ± 0.02 c	0.98 ± 0.00 e	2.40 ± 0.00 b
Sucrose	0.32 ± 0.02 d	0.32 ± 0.01 d	0.36 ± 0.02 c	0.23 ± 0.00 e	0.50 ± 0.01 b	0.34 ± 0.01 cd	0.57 ± 0.02 a
Total	8.10 ± 0.03 g	8.17 ± 0.04 f	11.13 ± 0.03 e	14.40 ± 0.03 b	13.56 ± 0.02 c	12.48 ± 0.00 d	14.85 ± 0.03 a
Organic acid content (g/100 g fw)							
Oxalic	0.01 ± 0.00 b	0.01 ± 0.00 b	0.01 ± 0.00 b	0.02 ± 0.01 b	0.20 ± 0.03 a	0.02 ± 0.01 b	0.02 ± 0.00 b
Citric	0.01 ± 0.00 b	0.01 ± 0.00 b	0.02 ± 0.00 b	0.03 ± 0.00 b	0.29 ± 0.03 a	0.02 ± 0.00 b	0.03 ± 0.00 b
Tartaric	0.01 ± 0.00 b	nd	nd	nd	nd	0.06 ± 0.01 a	nd
Malic	0.48 ± 0.01 d	0.51 ± 0.02 cd	0.52 ± 0.03 bc	0.56 ± 0.02 ab	0.57 ± 0.02 a	0.50 ± 0.02 cd	0.58 ± 0.03 a
Quinic	0.44 ± 0.01 d	0.44 ± 0.01 d	0.52 ± 0.01 bc	0.53 ± 0.03 b	0.48 ± 0.06 bcd	0.47 ± 0.04 cd	0.84 ± 0.01 a
Ascorbic	tr	tr	tr	tr	tr	tr	tr
Shikimic	tr	tr	tr	0.01 ± 0.00 a	0.02 ± 0.00 a	tr	tr
Fumaric	tr	tr	tr	tr	tr	tr	tr
Total	0.93 ± 0.01 e	0.97 ± 0.02 e	1.06 ± 0.06 d	1.14 ± 0.03 c	1.56 ± 0.00 a	1.06 ± 0.01 d	1.47 ± 0.02 b
Sugar/organic acids	8.62	8.34	10.40	12.52	8.64	11.66	10.03

Data are given as mean ± standard deviation (*n* = 3). nd: not detected. tr: traces. Mean values within a column with different letters (a–g) are significantly different (homogenous groups) at *p* ≤ 0.05.

**Table 3 foods-12-00105-t003:** Antioxidant activity of analysed apple juice and enriched apple smoothies.

Parameter	Sample						
	AJ	AJ+S01	AJ+S05	AJ+M5	AJ+F5	AJ+P5	AJ+C5
TP mg GAE/100 g fw	82.25 ± 2.36 f	91.19 ± f 6.78 ef	98.08 ± 7.79 e	132.20 ± 9.43 c	179.84 ± 9.95 b	194.06 ± 1.39 a	115.89 ± 4.57 d
CUPRAC mmol Fe^2+^/100 g fw	1.81 ± 0.09 f	4.77 ± 0.06 c	3.08 ± 0.04 e	3.61 ± 0.21 d	6.01 ± 0.04 b	7.04 ± 0.09 a	3.05 ± 0.02 e
FRAP mmol Trolox/100 g fw	0.58 ± 0.00 e	1.00 ± 0.01 d	0.65 ± 0.01 e	1.15 ± 0.04 c	1.77 ± 0.02 a	1.44 ± 0.01 b	0.95 ± 0.11 d
ORAC mmol Trolox/100 g fw	2.17 ± 0.17 c	2.23 ± 0.10 c	2.81 ± 0.19 b	3.07 ± 0.16 b	3.51 ± 0.23 a	2.96 ± 0.13 b	2.10 ± 0.15 c
DPPH^•^ mmol Trolox/100 g fw	0.47 ± 0.01 f	0.89 ± 0.03 c	0.50 ± 0.03 f	0.73 ± 0.04 d	1.26 ± 0.01 a	1.05 ± 0.04 b	0.60 ± 0.02 e
ABTS^•+^ mmol Trolox/100 g fw	0.48 ± 0.00 e	1.11 ± 0.01 c	0.57 ± 0.00 e	1.14 ±0.01 c	2.21 ± 0.01 a	1.82 ± 0.13 b	0.93 ± 0.03 d

Data are given as mean ± standard deviation (*n* = 3). Mean values within a column with different letters (a–f) are significantly different (homogenous groups) at *p* ≤ 0.05.

## Data Availability

All related data and methods are presented in this paper. Additional inquiries should be addressed to the corresponding author.

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
