# Peer review of "Effect of Apple Juice Enrichment with Selected Plant Materials: Focus on Bioactive Compounds and Antioxidant Activity"

_foods, 2022, doi:10.3390/foods12010105_

Round 1

Reviewer 1 Report

The manuscript entitled "Effect of Apple Juice Enrichment with Selected Plant Materials: Focus on Bioactive Compounds and Antioxidant Activity" is a good piece of work. However, there needs some clarifications and modification in the present form. Please find my comments and suggestions below;

1. The abstract is lacking background information and objectives. It is an important portion of the abstract

2. There is no quantitative data or statistical representations. It needs to be included in the abstract during revision

3. The introduction on apples are insufficient. Authors must include the global market trends of apples, nutritional content, pharmacological potentials etc in it. I suggest to include more literature on these aspects and elaborate the first paragraph

4. Authors could have included the data on the trace elemental composition of the apples (using AAS method)

5. The discussion section, especially on the antioxidant parameters could be discussed in relation with the incidence of degenerative diseases.

Author Response

  1. The abstract is lacking background information and objectives. It is an important portion of the abstract

We have slightly modified the Abstract following the Reviewer suggestion, but the 200 words limitation makes hard to improve it better.

  1. There is no quantitative data or statistical representations. It needs to be included in the abstract during revision

Following the Reviewer suggestion, quantitative data and statistical representations were added.

  1. The introduction on apples are insufficient. Authors must include the global market trends of apples, nutritional content, pharmacological potentials etc in it. I suggest to include more literature on these aspects and elaborate the first paragraph

The Introduction part was improved with information on the global market trends of apples, nutritional content, pharmacological potentials.

  1. Authors could have included the data on the trace elemental composition of the apples (using AAS method)

Trace elemental composition of the apples is a very important aspect of quality, from both toxicological and nutritional point of views, but we reputed that deepening also this aspect could create some mess in the structure of the manuscript. Indeed, the aim of the paper was to investigate the enrichment of apple juice with specific plant material containing antioxidant compounds, and for this reason analysis were selected to give proper information on this aspect of those new smoothies.

  1. The discussion section, especially on the antioxidant parameters could be discussed in relation with the incidence of degenerative diseases.

Following the Reviewer suggestion, this part was developed accordingly.

Reviewer 2 Report

The manuscript entitled "Effect of apple juice enrichment with selected plant materials:  focus on bioactive compounds and antioxidant activity" is good work that is scientifically sound and based on strong methodology.

My only concern about the manuscript is related to plant materials. In addition to their functional properties, are there any limitations to their use? The authors need to discuss the safety aspects of using these plant materials for fortification.

Please consider adding a picture of plant materials or prepared apple-enriched smoothies in the supplementary figure.

Other specific points:

In the first sentence in the Abstract, the authors used first scientific and then common plant names (lines 22-26). Please make it uniform and understandable.

Add reference(s) (Page 2, lines 44-47).

Instead of reference 5, it is preferable to cite the original reference (Page 2, line 49).

Lines 84-94: This part is too extensive. It is unnecessary to describe the experiment in detail.

More details on plant material collections should be provided

Add references for determining vitamin C, ash content, dry matter (Page 4, lines 157 and 163), and sensory tests (Page 5, line 201).

Author Response

My only concern about the manuscript is related to plant materials. In addition to their functional properties, are there any limitations to their use? The authors need to discuss the safety aspects of using these plant materials for fortification.

The plant materials used in this study were selected because their consumption is generally recognized safe. A brief discussion on the safety aspects of using the plant materials for fortification was added in the manuscript.

Please consider adding a picture of plant materials or prepared apple-enriched smoothies in the supplementary figure.

A new Figure S1 was added in the supplementary file reporting the plant material used for the investigation, the semi-finished products and the obtained final smoothies.

Other specific points:

In the first sentence in the Abstract, the authors used first scientific and then common plant names (lines 22-26). Please make it uniform and understandable.

The common plant names were replaced by scientific names to make the Abstract uniform.

Add reference(s) (Page 2, lines 44-47).

As suggested by Referee, new references were added to the manuscript.

Instead of reference 5, it is preferable to cite the original reference (Page 2, line 49).

As suggested by Referee, the “Nour, 2022” was replaced by the original reference (Sulaiman 2017).

Lines 84-94: This part is too extensive. It is unnecessary to describe the experiment in detail.

As suggested by Referee, the part was modified and shortened.

More details on plant material collections should be provided

More information was added on plant material collections.

Add references for determining vitamin C, ash content, dry matter (Page 4, lines 157 and 163), and sensory tests (Page 5, line 201).

As requested by Referee, the references were added.

Round 2

Reviewer 1 Report

Authors have made significant revision of their manuscript as per the comments.